# Identification of Clinician Training Techniques as an Implementation Strategy to Improve Maternal Health: A Scoping Review

**DOI:** 10.3390/ijerph20116003

**Published:** 2023-05-30

**Authors:** Cheryl A. Vamos, Tara R. Foti, Estefanny Reyes Martinez, Zoe Pointer, Linda A. Detman, William M. Sappenfield

**Affiliations:** 1USF’s Center of Excellence in Maternal and Child Health Education, Science & Practice, The Chiles Center, College of Public Health, University of South Florida, Tampa, FL 33612, USA; 2College of Public Health, University of South Florida, Tampa, FL 33612, USA; tfoti@usf.edu (T.R.F.); zpointer@usf.edu (Z.P.); 3College of Public Health, Florida Perinatal Quality Collaborative, University of South Florida, Tampa, FL 33612, USA; estefanny@usf.edu; 4The Chiles Center, College of Public Health, Florida Perinatal Quality Collaborative, University of South Florida, Tampa, FL 33612, USA; ldetman@usf.edu (L.A.D.); wsappenf@usf.edu (W.M.S.)

**Keywords:** clinician education, clinician training, quality improvement, implementation science, maternal health, perinatal health

## Abstract

Training is a key implementation strategy used in healthcare settings. This study aimed to identify a range of clinician training techniques that facilitate guideline implementation, promote clinician behavior change, optimize clinical outcomes, and address implicit biases to promote high-quality maternal and child health (MCH) care. A scoping review was conducted within PubMed, CINAHL, PsycInfo, and Cochrane databases using iterative searches related to (provider OR clinician) AND (education OR training). A total of 152 articles met the inclusion/exclusion criteria. The training involved multiple clinician types (e.g., physicians, nurses) and was predominantly implemented in hospitals (63%). Topics focused on maternal/fetal morbidity/mortality (26%), teamwork and communication (14%), and screening, assessment, and testing (12%). Common techniques included didactic (65%), simulation (39%), hands-on (e.g., scenario, role play) (28%), and discussion (27%). Under half (42%) of the reported training was based on guidelines or evidence-based practices. A minority of articles reported evaluating change in clinician knowledge (39%), confidence (37%), or clinical outcomes (31%). A secondary review identified 22 articles related to implicit bias training, which used other reflective approaches (e.g., implicit bias tests, role play, and patient observations). Although many training techniques were identified, future research is needed to ascertain the most effective training techniques, ultimately improving patient-centered care and outcomes.

## 1. Introduction

Infant and maternal morbidity and mortality rates improved considerably over the past several decades; however, that improvement has been inconsistent with challenges such as preterm birth and infant death rates still high or rising in the United States, as well as other alarming maternal issues such as opioid use, where the number of drug-related pregnancy-associated deaths has more than doubled in recent decades [1,2,3,4]. Clear racial and ethnic disparities in maternal and infant morbidity and mortality have been documented in the United States [5,6]. Non-Hispanic Black women face a mortality rate three to four times higher than their Non-Hispanic White counterparts, and other racial/ethnic minorities face similarly elevated pregnancy-related mortality levels [5,7,8]. As for infant health outcomes, Black infants have a mortality rate 2.2-fold higher than White infants, a value that has considerably increased over the last 50 years [6]. 

These complexities are exacerbated by access to and delivery of poor-quality healthcare [9,10,11]. Additionally, implicit bias, the “thoughts and feelings that often exist outside of conscious awareness” [12] that can unknowingly affect human understanding, actions, and decisions, affects clinicians’ decision making and patient interaction skills as stereotypes about certain groups begin to unconsciously affect the quality of care delivered to patients, particularly with opioid users, those economically challenged, and racial/ethnic minorities [12,13]. When working with and caring for individuals from varying backgrounds, assuring cultural competency that reflects and respects the diversity of patients based on their behaviors, race, religion, beliefs, and views is critical to delivering compassionate and patient-centered care [14]. 

The underlying system-level factors influencing the determinants of health, disease progression, and quality of healthcare are complex. The burden of these challenges and disparities, including those brought about by implicit biases, may be alleviated with the introduction of provider training and education to ensure clinicians have the knowledge, skills, and resources necessary to deliver evidence-based, patient-centered, and equitable care rooted in cultural humility and shared decision making [7,13,15]. Provider training and education has been identified as a key strategy during the implementation of healthcare improvements and innovations [16]. Provider training is also a common strategy used during perinatal quality improvement initiatives across a diverse range of settings and topic foci [17,18]. 

As noted by Siegl et al. [19], professional development and education activities should be designed to support the implementation of clinical guidelines and protocols—which are often continuously updated due to scientific advancements—to facilitate the administration of evidence-based, high-quality care. However, barriers to the implementation of provider training are plentiful, including the complexity of perinatal outcome improvement; competing clinical demands; lack of provider time; lack of staff buy-in; lack of structured quality improvement curricula; lack of standardized, evidence-based guidelines and best practices; and lack of provider understanding of quality improvement methodologies and training techniques [20,21]. 

Presently, a range of educational strategies used within maternal and child health (MCH) clinical settings have been documented, including simulation, didactic lectures, discussion groups, Grand Rounds, videos, train-the-trainer, and more [22,23,24,25,26]. Such strategies have reached interdisciplinary provider groups, including obstetricians and gynecologists, midwives, nurses, lactation consultants, and various other professionals [27,28]. Though a spectrum of training techniques is sometimes referenced in the literature, there still exist significant gaps regarding a complete identification of these techniques, as well as an understanding of their impact. Furthermore, since the use of each education strategy and its corresponding environment is highly specific and individualized, it is unclear how broadly the utility of these educational strategies can be inferred.

A significant lack of clarity exists surrounding the conceptual identification and evaluation of provider training techniques used specifically for healthcare improvement, including MCH services, resulting in a culture of constantly shifting understandings of what techniques qualify as best practices for clinicians across various professional backgrounds [16]. Subsequently, it can be difficult to plan and deliver new quality improvement projects in MCH across clinical settings with limited understanding regarding the theoretical and practical utility of provider education and the respective training techniques. 

Therefore, the purpose of this scoping literature review was to identify a range of reported provider education and training techniques across clinical settings that facilitate the implementation of guidelines, promote clinician behavior change, optimize clinical outcomes, and encourage reflection on implicit biases to promote culturally competent care within MCH. Such understanding could be used to inform future perinatal quality improvement projects and potentially identify best practices for clinician education to promote improved clinical outcomes for all patient groups. The comprehensive list of strategies, in addition to information collected about the groups and settings in which such methods are most appropriately utilized, will help fill the existing gap in knowledge pertaining to best practices for supporting various quality improvement initiatives. Given the persistent disparities across MCH outcomes, a secondary aim was to determine whether additional reported provider education and training techniques are reported within the literature specific to cultural competence and implicit bias.

## 2. Materials and Methods

Scoping reviews are the preferred methodology in cases addressing broad topics with a variety of study designs, generally without highly specific research questions and without quality assessment of the identified studies [29]. Since our research question was broadly defined in an effort to identify a range of educational strategies and related training characteristics used within MCH, a scoping review was the selected methodology. 

### 2.1. Primary Search Strategy

The following databases were searched from December 2019 through February 2020: PubMed, CINAHL, PsycInfo, and Cochrane. Hand searching was performed on national quality improvement organizational websites that focus on perinatal, maternal, and child health, as well as the identified CDC-funded perinatal quality collaborative websites. Additional database searches were performed to identify related articles using “de-implementation” as the keyword. After the identification of articles through database screening, titles were reviewed based on inclusion and exclusion criteria. Articles with included titles were downloaded into EndNote software, and duplicates were excluded. Abstract reviews, followed by full-text reviews, took place. Articles that were identified following full-text review were screened by two reviewers, with the lead author serving as a tiebreaker in the case of any reviewer disagreement. A manual review of references of review articles that otherwise met inclusion criteria was performed to identify additional related literature.

### 2.2. Search Terms

Search terms were optimized by database but started with a combination of (provider OR clinician) AND (training OR education) (see Table 1).

### 2.3. Inclusion and Exclusion Criteria

Titles were selected based on the inclusion criteria: (1) primary focus was on reproductive, pregnancy-associated, and/or perinatal health, including the neonatal period; (2) an explicit implementation of education or training in the workplace, either as a stand-alone or as a component of a quality improvement or another program; (3) clinicians as the main target population of the education or training; and (4) published between 2010 and 2019 in English. Articles were excluded based on the following: (1) the education or training was implemented in a developing or low-resource nation based on the United Nations classification [30]; (2) clinician students were the primary audience; or (3) articles were based on a systematic or scoping review or were a review protocol.

### 2.4. Selection Procedure

A flowchart of the selection procedure is shown in Figure 1 (selection procedure flow diagram). Articles (n = 13,907) were identified through database searches and were screened by one of the team members (TF). After excluding 13,099 titles and 74 duplicates, 734 abstracts were reviewed, of which 247 were excluded. Full-text articles (n = 487) were assessed, with 362 excluded. The reference sections of articles were screened to identify other potential literature to include in the dataset, which resulted in 12 additional articles. The remaining articles were reviewed by two team members [ER and ZP], and tiebreakers were determined by the lead author. The final total included in this scoping review is 152 articles. Hand searches of MCH QI-related organizations and perinatal quality collaborative websites, as well as the de-implementation database searches, did not result in any included articles.

### 2.5. Data Abstraction

The final set of included articles was input into a shared spreadsheet. Abstraction fields were added to the spreadsheet, including information relating to the article, study location, targeted provider types, training topic, education strategy(ies), related guidelines or evidence-based practice, the inclusion of cultural competence or patient–provider rapport or communication, setting, educational assessment, and strengths and weaknesses of the strategy and overall study. The full list of abstraction fields is listed in Table 2. Three team members independently abstracted study elements into the spreadsheet. Team members met weekly to further refine abstraction and inclusion/exclusion criteria. 

### 2.6. Secondary Search Strategy

A search of the literature was performed to better understand provider education strategies and approaches used to uncover and address implicit bias. In this ad hoc review, multiple searches were performed in PubMed using the following search terms: (“cultural competenc*” or “implicit bias”) and provider and (training or education); (“cultural competenc*” or “implicit bias”) and (provider or clinician or doctor or physician or nurse) and (training or education) and (opioid or drug or addiction or behavioral health); (“cultural competenc*” or “implicit bias”) and (provider or clinician or doctor or physician or nurse) and (training or education) and (reproduct* or perinat* or puerpere* or postpart* or women* or matern* or child)); and “implicit bias” and training and clinician and (“maternal health” or “child health” or “perinatal health” or “maternal*child health”). The searches were limited to the past 10 years (2011–2020) and were published in English in peer-reviewed journals. Search result titles were reviewed and downloaded into EndNote if they were related to an overview of implicit bias or provider training strategies. 

## 3. Results

A total of 152 articles were included in the final data set after the inclusion and exclusion criteria were applied. Findings below report on training participant demographics, training characteristics, adherence to guidelines/evidence-based practice, outcome assessments, and the incorporation of training on interpersonal communication. 

### 3.1. Participants and Demographics

Education initiatives were directed at various types of clinicians. The following clinicians were identified in the articles (percentages are indicative of the percent of studies that reported the specific provider type): nurses (54.7%), midwives (43.1%), OB/GYN (39.4%), MD (not OB/GYN or pediatrician) (36.5%), other participants (32.1%), advanced practice clinicians (25.6%), and pediatricians (9.5%). Of note, most education initiatives were directed toward multiple clinicians. For example, simulations could consist of teams made up of nurses, physicians, and other advanced practice clinicians. Other educational initiatives showed similar findings with a multi-professional approach. 

Training factors related to duration and location were abstracted. The length of training varied between initiatives, with training ranging from less than 30 min long (5.1%), ½ hour to 1 day (40.1%), and at least 8 h (32.1%). Approximately one-quarter (22.6%) of training activities did not disclose the total training length. Training activities were implemented across a wide range of settings, including hospitals (38.0%), academic medical centers (24.8%), off-site locations (17.5%), and clinics (10.9%). 

### 3.2. Training Characteristics

Articles from various life stages were used to identify prominent training techniques. Life stages across the reproductive lifecycle included labor and delivery (34.3%, n = 47), pregnancy (35.0%, n = 48), woman (non-pregnant) (23.4%, n = 32), postpartum (15.3%, n = 21), and neonatal/infant (11.7%, n = 16). 

Similar training topics were grouped together to identify the most prominent training topics within the abstracted articles (articles were assigned to only one group). The following topics were identified: maternal/fetal morbidity and mortality (26%, n = 39), teamwork and communication (14%, n = 21), screening, assessment, and testing (12%, n = 18), breastfeeding and skin-to-skin (8%, n = 12), trauma and abuse (8%, n = 12), neonatal care and resuscitation (7%, n = 10), substance abuse (4%, n = 6), contraception and sterilization (4%, n = 6), perinatal loss and abortion (4%, n = 6), perinatal mental health (4%, n = 6), work-related (3%, n = 4), women’s health (2%, n = 3), fertility and sexual/reproductive health (2%, n = 3), healthy weight (2%, n = 3), and labor and postpartum support (2%, n = 3). 

Most studies used multiple training techniques. The top five most common training techniques identified were didactic (65.0%), simulations (38.7%), hands-on (scenario, role play, or demonstration) (27.7%), online training (24.8%), and discussion (27.0%). Identified training techniques, compiled definitions, and related articles can be found in Table 3.

Other materials to support training techniques were also identified within the articles and used as supplementary training resources and opportunities, including the following: paper materials (e.g., textbooks, packets/workbooks/booklets, learner portfolios, manuals, fact sheets, brochures, and pocket guides/wallet cards); digital materials (e.g., CD-ROMs, DVDs, websites, electronic reminders, toolkits, module-based learning, and videos); promotional materials (e.g., posters, flyers, buttons, notepads, email newsletters, and desktop calendars); visuals (e.g., photographs, X-rays, graphs/charts, and cue cards); opportunities for skill building and professional development (e.g., patient rounding, practical skills training sessions, case studies, training certificates, and pledges of action); and opportunities for evaluation (e.g., feedback, overviews of guidelines/protocols, competency evaluations, and support for training attendees).

### 3.3. Guidelines/Evidence-Based Practice

Articles were evaluated to determine if the training techniques were informed by guidelines or evidence-based practice within the fields of women’s reproduction, pregnancy, and perinatal care. Within the articles, only 42.3% of articles explicitly referenced that training was being used to help facilitate guideline implementation or evidence-based practice. The most referenced organizations within these articles were the Association of Women’s Health, Obstetric and Neonatal Nurses (AWHONN), the American College of Obstetricians and Gynecologists (ACOG), and the American Academy of Pediatrics (AAP). 

### 3.4. Assessments

Articles were examined to evaluate whether changes in knowledge, confidence, or clinical outcome were assessed after the implementation of the training techniques. Based on the articles that were abstracted, only 38.7% assessed knowledge, and 37.2% assessed confidence. When examining the articles that assessed both knowledge and confidence, 22.6% assessed both of these measures. Only 30.7% of articles reported assessing a clinical outcome after implementing the training technique.

### 3.5. Interpersonal Communication (Provider-Provider and Provider-Patient) 

Articles were also examined to determine if the training technique addressed cultural competency/sensitivity. However, cultural competency/sensitivity within these articles primarily focused on interprofessional teamwork and communication and patient–provider communication and support. Only one-fifth of the articles (22%, n = 30) included cultural competency/sensitivity within their training techniques. Within patient–provider communication and support, there was a focus on topics such as bereavement/delivering bad news, weight, vaccines, sexual orientation, intimate partner violence, female genital mutilation, breastfeeding, trauma, abortion, substance abuse, IUDs/implants, and other general topics pertaining to cultural competency/sensitivity (i.e., communication, non-verbal communication, non-judgmental views, attitudes, and compassion). 

### 3.6. Implicit Bias Search Results

A total of 207 articles were reviewed to evaluate article relevance by title and abstract review. Twenty-two articles addressed the implementation of provider training related to implicit bias within the workplace. Additional strategies identified in the implicit bias literature review that were not identified in the general provider training strategies literature review are listed by the most commonly reported strategy in Table 4. In addition, while demonstration and role playing were identified as strategies in the primary review, the more specific strategies of role modeling in clinical settings, observed patient encounters with faculty feedback, and role play with scripted cases were identified in the implicit bias literature.

## 4. Discussion

This scoping review aimed to identify a range of reported provider education and training techniques across clinical settings that promote perinatal clinician behavior change, optimize clinical outcomes, and encourage reflection on implicit biases to promote culturally competent care. Of the articles included, the major provider types of focus were nurses, midwives, and OB/GYNs (although many studies focused on multiple or interdisciplinary provider groups). They featured techniques that were delivered in hospitals or academic medical centers, lasted between a half-day and a full day in length, targeted labor and delivery or pregnancy stages, and aimed to address maternal/infant morbidity and mortality. 

The most frequent techniques identified included didactic, followed by simulations, hands-on (scenario, role play, or demonstration), online training, and discussion. Other materials were also identified as supplementary training resources (e.g., paper/digital materials and visuals) and opportunities for skill building and professional development (e.g., training certificates). The use of applied learning strategies (e.g., simulations; role play) is important as it is well known that best practices in adult pedagogy underscore the value of using combined training strategies and optimizing long-term behavior change through active learning techniques [177,178]. Furthermore, learning through application and case studies is a common approach utilized in health professional training programs [179,180,181]. 

Unfortunately, less than half of the articles referenced training on guidelines or evidence-based practices. Although this information could have been omitted in the articles included in this scoping review, training based on the current evidence related to that topic is critical. Various frameworks on healthcare implementation and guideline adherence identify factors such as evidence strength and quality [182], along with familiarity, awareness, and agreement with guidelines [183], as important determinants influencing acceptance and adoption of a new health service or practice, ultimately impacting clinician and organization behavior change. Thus, more integration of dissemination and implementation of scientific principles and methods [184,185,186,187] within provider education across MCH topics could support and assist with understanding which techniques work best across topics, settings, and other contextual factors.

In addition, only about one-third of the articles assessed the impact of the technique on knowledge, confidence, or clinical outcome. This is concerning, as considerable time, effort, and emphasis is placed on provider education and training as a key strategy to improve the quality of care and health outcomes [188,189,190,191,192,193]. This implies that most studies employing provider training techniques are not explicitly evaluating this implementation strategy. The lack of conducting and disseminating evaluation activities has led to a considerable gap in research and practice on provider education techniques across MCH interventions. Future research should assess implementation outcomes (e.g., acceptability, adoption, feasibility, and costs) [194] related to the training technique employed to contribute to the literature regarding provider training as a key implementation strategy. 

Furthermore, this scoping review identified a lack of articles (approximately one-fifth) that included cultural competency/sensitivity to support interpersonal communication (provider-provider and provider-patient) within their training techniques. There are widespread national movements, calls for action, and political will to address MCH disparities and improve health equity [193,195,196,197,198,199]. Therefore, it is surprising that more articles included in this review over the study time period (2010–2019) did not reference a training element specific to cultural competency and/or patient-centered care. Although such training may be delivered within an organization as part of employee training requirements, it remains unknown which provider training techniques are utilized and effective in practice for this critical determinant of healthcare access, quality, and outcomes, as is noted in key MCH and quality improvement efforts, as well as being a highly popular saying, ‘what gets measured gets done/managed’. Thus, reporting on the development, implementation, and evaluation of training activities across interpersonal and quality care attributes such as cultural competency and implicit bias is essential to understand not only effective training strategies, but to document commitment and change in this area. Moreover, of those articles that explicitly employed provider training techniques to address implicit bias, the majority utilized role play and patient encounter observations, highlighting the value of interactive and experiential training on this topic.

Findings must be considered in light of the noted limitations. First, not all of the articles included in this review provided comprehensive detail about the training techniques and their implementation. For instance, articles often frequently omitted or failed to define or operationalize training techniques, including the length of training, implementation locations, participating provider groups, and methods and results related to evaluating changes in knowledge, attitude, and clinical improvement. Thus, when abstracting, categorizing, and sorting information across data fields, not all data categories and fields may be mutually exclusive or accurately represent training parameters. Furthermore, the exclusion criteria employed limits generalizability. Articles were excluded if they were not published in English and/or if they targeted student learners. This limits study findings, as valuable research may be published among health professional student training programs, which could aid in understanding best practices within a provider training context. Lastly, inherent in studies that employ a review methodology, there is a time lag from when the study was conducted to publication, and thus it is possible that there may be additional literature published on this topic in the past couple of years. However, given the voluminous number of training techniques identified in this review, as well as the key finding that a minority of articles reported evaluating change in clinician knowledge, confidence, or clinical outcomes, it is highly likely that a similar finding of a significant gap in this area and the need for rigorous evaluations on such techniques to assist in improving healthcare quality and outcomes would remain.

Nonetheless, there were many strengths to this study. During the scoping review methodology, investigators were able to identify a broad range of techniques and training characteristics, placing the findings into a comprehensive list that did not previously exist in the published literature. By creating standardized definitions for the purposes of article categorization, it became easier to identify which training techniques and training characteristics are most commonly implemented.

## 5. Conclusions

In conclusion, this scoping review identified a number of training techniques and characteristics within the field of MCH. Future research is needed to identify the most effective training techniques to increase clinicians’ knowledge and confidence, and subsequently improve patient-centered and healthcare delivery outcomes. Since health disparities persist within MCH, training techniques should also include communication skills and cultural competency components. Additional research and practice efforts on providing training as a key implementation strategy are needed to advance clinical healthcare and MCH outcomes.

## Figures and Tables

**Figure 1 ijerph-20-06003-f001:**
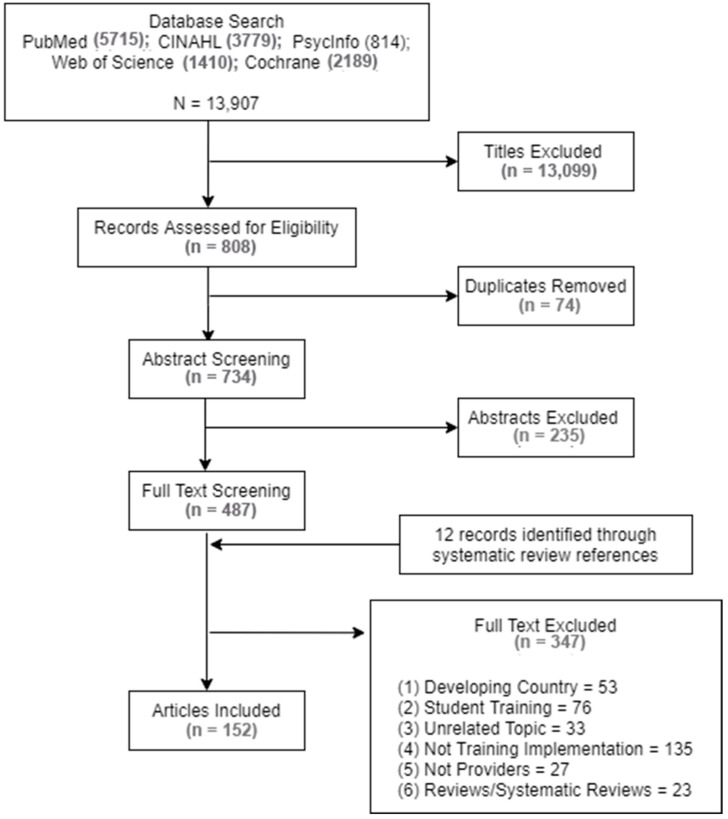
Selection procedure flow diagram.

**Table 1 ijerph-20-06003-t001:** Search terms by database.

Database	Search Terms
PubMed	(educat* or train* teach* or workshop or simulation or “continuing medical education” or “quality improvement”) AND (provider or physician or clinician or nurse or midwife or obstetrician or practitioner or health professional) AND (perinat* or obstetric* or gynecolog* or pregnan* or matern* or women* or prenatal or postnatal or postpartum or puer* or reproduct*)
CINAHL	(educat* or train* teach* or workshop or simulation or “continuing medical education” or “quality improvement”) AND (provider or physician or clinician or nurse or midwife or obstetrician or practitioner or health professional) AND (perinat* or obstetric* or gynecolog* or pregnan* or matern* or women* or prenatal or postnatal or postpartum or puer* or reproduct*) AND subject (nursing skills OR nursing knowledge OR midwives education OR obstetric nursing education OR clinical competence OR nurses education OR childbirth OR professional role OR postnatal care OR professional knowledge OR nursing staff, hospital education OR physicians education OR quality improvement OR women’s health OR postnatal period OR postnatal care OR obstetrics education OR gynecology education OR learning environment OR clinical evaluation OR education, interdisciplinary OR lactation education OR breast feeding OR multidisciplinary care team education OR prenatal care OR pregnancy OR health screening OR health personnel education OR perinatal care education OR teaching methods OR professional knowledge evaluation OR seminars and workshops OR communication skills OR program evaluation OR simulations)
PsycInfo	AB (educat* or train* teach* or workshop or simulation or “continuing medical education” or “quality improvement”) AND AB (provider or physician or clinician or nurse or midwife or obstetrician or practitioner or health professional) AND AB (perinat* or obstetric* or gynecolog* or pregnan* or matern* or women* or prenatal or postnatal or postpartum or puer* or reproduct*) AND SU (education or medical education or quality of care OR program evaluation OR teaching)
Cochrane	(clinician or provider or physician or doctor or nurse or midwife) AND (training or education or evaluation or implementation) in Title Abstract Keyword

Notes: * Truncation symbol to include alternative word endings in search results. AB-PsycInfo indicator to include articles with the indicated words or phrases located within the abstract in search results. SU-PsycInfo indicator to include articles with the indicated words or phrases located within the subject in search results.

**Table 2 ijerph-20-06003-t002:** Data abstraction fields.

Data Category	Data Fields
Location	Country/State of Study
Clinicians	Types of Clinicians
Training Topic	Topic Topic Group Life Stage Addressed
Training Strategies	Type of Training Training Duration Training Description Training Goal(s) Description of Cultural Competence/Communication Training Setting
Patient–Provider Relational Skills	Inclusion of Patient–Provider Relational skills (communication, bias, cultural competence, etc.)
Guidelines/Evidence-Based Practice	Training Based on Guidelines/EBP Link to or Citation for Guideline/EBP
Assessment	Methods Analysis Measurement Assessment of Learning (knowledge) Assessment of Confidence Assessment of Clinical Change or Outcome Findings
Strengths and Weaknesses	Strengths Weaknesses Other Notes

Note: EBP = Evidence-based practice.

**Table 3 ijerph-20-06003-t003:** Training techniques, definitions, and articles reporting the use of each technique.

Training Technique	Definition	Articles Reporting Use of Technique
Didactic (n = 96)	Instruction using a trainer (may be informal); traditional lecture format.	[19,24,26,27,28,31,32,33,34,35,36,37,38,39,40,41,42,43,44,45,46,47,48,49,50,51,52,53,54,55,56,57,58,59,60,61,62,63,64,65,66,67,68,69,70,71,72,73,74,75,76,77,78,79,80,81,82,83,84,85,86,87,88,89,90,91,92,93,94,95,96,97,98,99,100,101,102,103,104,105,106,107,108,109,110,111,112,113,114]
Simulation (n = 59)	A representation of real-world healthcare scenarios in an educational environment, often using simulator mannequins.	[25,27,28,48,49,52,57,59,65,66,67,75,82,83,86,87,88,91,92,96,97,98,99,100,104,106,109,111,112,113,115,116,117,118,119,120,121,122,123,124,125,126,127,128,129,130,131,132,133,134,135,136,137]
Hands-On (Scenario, Role Play, or Demonstration) (n = 43)	Hands-on demonstration by trainer or activity by participants; may include role play of various clinical scenarios.	[26,31,35,41,43,44,46,50,51,56,57,60,61,62,64,65,68,70,71,73,76,82,84,90,92,94,95,102,103,104,111,112,113,116,138,139,140]
Discussion (n = 37)	Formal discussion within training amongst participants or in a group with the facilitator; includes Question and Answer sessions.	[19,24,31,32,34,36,39,40,41,43,44,51,53,57,60,68,76,79,81,83,84,85,86,91,93,95,101,104,112,113,115,116,119,139,141,142,143]
Online (n = 34)	Training occurred through the internet, or the participants did not need to all be present in the same physical space due to use of technology that enabled remote participation.	[19,24,25,32,39,41,42,45,55,63,68,77,89,94,105,111,116,119,127,138,140,141,143,144,145,146,147,148,149,150,151,152,153,154]
Simulation Debrief (n = 34)	A debrief session among simulation participants following the formal simulation.	[27,48,49,52,59,83,87,88,92,96,97,98,100,104,111,112,113,115,117,119,121,122,123,124,125,126,127,128,129,130,131,132,134,135]
Video (n = 33)	The training incorporated the use of video in some way.	[24,34,35,38,41,52,53,65,73,82,88,92,96,98,102,103,104,107,111,113,116,117,123,127,132,138,139,144,146,147,149,150,154]
Published Curriculum (n = 32)	Use of a curriculum that is widely available, such as on the internet, through purchase, or in print.	[19,25,28,32,33,39,44,45,53,56,64,65,67,74,77,78,81,85,88,107,109,111,114,132,138,139,141,142,148,149,154,155]
Continuing education credits (n = 18)	Providing continuing education credits, such as Continuing Medical Education (CME); are often required to maintain professional certifications.	[19,32,36,37,40,41,85,92,102,103,104,105,138,139,149,151,152,153]
Train the Trainer (n = 15)	A training strategy in which a group of employees is trained to facilitate learning among other employees.	[26,27,28,31,35,39,50,65,74,83,101,111,115,147,156]
Mentor/Supervisor (n = 10)	The assignment of a mentor or supervisor to provide or oversee training or clinical practice.	[32,46,66,90,99,104,106,140,143,157]
One-on-One (n = 6)	A trainer meeting with a participant individually to provide training.	[24,32,68,84,108,158]
Grand Rounds (n = 4)	A type of medical education in which a particular audience of clinicians gather to discuss medical problems and treatments.	[19,24,25,111]
Journal Club (n = 2)	A group of people meeting to discuss recent articles in academic literature.	[24,32]
Lunch and Learn (n = 1)	Voluntary meetings, training sessions, or presentations that take place during lunch.	[38]

**Table 4 ijerph-20-06003-t004:** Additional training techniques identified in implicit bias literature.

Training Technique	Articles Reporting Use or Explanation of Technique
Assigned reading	[159]
Case presentations/case studies/clinical vignettes	[159,160,161,162,163,164]
Clinical encounter/patient vignette videos	[162,165,166]
Commitment to act or change	[159]
Debriefing and facilitation/discussion	[159,161,166,167,168]
Experiential, group, and/or interactive exercises and activities	[167,169,170,171]
Health equity rounds	[160]
Feedback on patient encounters	[159,162]
Facilitated and self-reflection	[159,167,168,169,172,173]
Mindfulness meditation	[174]
Narrative and reflective writing	[159]
Participate in community/cultural needs assessment	[159,168]
Present relevant historical context	[160]
Presentations from community groups	[175]
Provide advocacy tools	[160]
Roleplaying and role modeling	[159,169,172]
Self-cultural reflection	[171]
Service learning	[176]
Storytelling	[172]
Toolkit	[166]

## Data Availability

The datasets used and/or analyzed during the current study are available from the corresponding author upon reasonable request.

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
