# Peer review of "Identification of Clinician Training Techniques as an Implementation Strategy to Improve Maternal Health: A Scoping Review"

_ijerph, 2023, doi:10.3390/ijerph20116003_

Round 1

Reviewer 1 Report

I do appreciate the choice of a scoping review as methodology to approach the research question.

Few concerns:

1. Line 115: why did you ended the search on databases in February 2020? I believe that in the last 2 years more evidence could be extracted from available literature;

2. Quality of Figure 1 should be improved;

3. Line 176: same concern as point 1;

4. Line 329: citation(s) missing;

5. Line 344: please briefly summarize your findings.

Reviewer 2 Report

This manuscript describes the contents of a scoping review of Clinician Training Techniques. It's an interesting subject and easy to read, but I have a comment for the author.

Comment: I don't quite understand the definition of training. There are various words such as “training”, “education”, “leaning”, “practice”, but what does “training” in this paper refer to, including the relationship between these? Also, you used "educat* or train*" when searching the database, but does that mean education = training? Or is "training" included in "education"?

Table 3 classifies training techniques. For example, "Didactic", "Simulation", "Discussion", etc. are listed as the elements, but is it appropriate to classify them as equivalent factors? Many of these factors are performed as a combination (for example; “Simulation” and “Simulation Debrief” or “Discussion”). I think Table 3 could be organized a little more. Also, "Journal Club" and "Lunch and Learn" are listed as items, but are they also treated as training?

Round 2

Reviewer 2 Report

Thank you for your reply to my comments.